# Carfilzomib Promotes the Unfolded Protein Response and Apoptosis in Cetuximab-Resistant Colorectal Cancer

**DOI:** 10.3390/ijms22137114

**Published:** 2021-07-01

**Authors:** Ahmad Zulkifli, Fiona H. Tan, Zammam Areeb, Sarah F. Stuart, Juliana Gomez, Lucia Paradiso, Rodney B. Luwor

**Affiliations:** 1Department of Surgery, The University of Melbourne, The Royal Melbourne Hospital, Clinical Sciences Building, Parkville, VIC 3050, Australia; a.zulkifli@student.unimelb.edu.au (A.Z.); fionat@student.unimelb.edu.au (F.H.T.); z.areeb@student.unimelb.edu.au (Z.A.); sstuart@student.unimelb.edu.au (S.F.S.); JulianaG@student.unimelb.edu.au (J.G.); lucia.paradiso@hotmail.com (L.P.); 2Fiona Elsey Cancer Research Institute, Federation University Australia, Ballarat, VIC 3350, Australia

**Keywords:** EGFR, carfilzomib and unfolded protein response

## Abstract

Cetuximab is a common treatment option for patients with wild-type K-Ras colorectal carcinoma. However, patients often display intrinsic resistance or acquire resistance to cetuximab following treatment. Here we generate two human CRC cells with acquired resistance to cetuximab that are derived from cetuximab-sensitive parental cell lines. These cetuximab-resistant cells display greater in vitro proliferation, colony formation and migration, and in vivo tumour growth compared with their parental counterparts. To evaluate potential alternative therapeutics to cetuximab-acquired-resistant cells, we tested the efficacy of 38 current FDA-approved agents against our cetuximab-acquired-resistant clones. We identified carfilzomib, a selective proteosome inhibitor to be most effective against our cell lines. Carfilzomib displayed potent antiproliferative effects, induced the unfolded protein response as determined by enhanced CHOP expression and ATF6 activity, and enhanced apoptosis as determined by enhanced caspase-3/7 activity. Overall, our results indicate a potentially novel indication for carfilzomib: that of a potential alternative agent to treat cetuximab-resistant colorectal cancer.

## 1. Introduction

Metastatic colorectal cancer (mCRC) is one of the highest causes of cancer mortality [1,2]. First- and second-line therapy in the metastatic setting consists of fluoropyrimidine-based chemotherapy [3,4,5], while targeted therapies such as the epidermal growth factor receptor (EGFR) inhibitors cetuximab and panitumumab and the vascular endothelial growth factor inhibitor bevacizumab are now also routinely used [6,7,8]. However, despite these agents producing improvements in patient outcomes, treatment failure is frequently observed and 5-year survival rates for patients with metastatic disease remain below 15% [9,10]. Cetuximab has shown promise in mCRC treatment. However, patients who initially responded well to cetuximab often develop or acquire resistance [11]. Mutations in the Kirsten-Ras (K-RAS) gene (present in 30–40% of mCRC) are currently the strongest predictive marker of resistance to EGFR-targeted therapy [11,12]. Indeed, 90% of mCRC patients harbouring K-RAS mutations show no therapeutic benefit to cetuximab or panitumumab. Due to the lack of response, the American Society of Clinical Oncology and the European Medicines Agency have issued guidelines to screen patient biopsies for K-RAS mutations prior to treatment [13,14] and subsequently to administer only EGFR-based agents into patients with tumours expressing wild-type (wt) K-RAS. However, over 40–60% of patients who express wild-type K-RAS still respond poorly to cetuximab, suggesting resistance through other mechanisms [13,15,16]. Therefore, a deeper understanding of those resistance mechanisms is needed and more importantly continued evaluation for improved therapeutic agents is still essential.

One potential target for treatment is the proteasome. The proteasome is a complex of proteases that processes proteins that have been targeted for degradation [17]. Multiple cellular activities depend on this process, and inhibition leads to accumulation of aberrated proteins and eventually cell death [18]. The success of first- and second-generation proteasome inhibitors such as bortezomib and carfilzomib has produced promising results in clinical trials [19,20]. Carfilzomib has been approved by the FDA for treatment of recurrent multiple myeloma [21]. Compared with bortezomib, carfilzomib has lower toxicity as well as minimal off-target effects [22]. Carfilzomib showed significant single-agent activity against two colorectal cancer cell lines [23]. However, these two cell lines harbour K-RAS mutations [24]. Similarly, others have shown carfilzomib efficacy against colorectal cancer cells with BRAF mutations [25,26]. However, little is known about whether carfilzomib can inhibit CRC cells with wt K-RAS expression and whether it can overcome cetuximab resistance to these cells.

One mechanism for initiating apoptosis in cancer cells by proteosome inhibitors is to induce endoplasmic reticulum (ER) stress through the accumulation of mis-folded proteins [27]. Under pathophysiological conditions or when cells are challenged with inhibitors, the ER is unable to manage appropriate protein folding, resulting in the induction of ER stress [28,29]. The unfolded protein response (UPR) is a collection of adaptive signalling pathways that sense ER stress and either promote cell survival, or, when ER stress is too severe or prolonged, trigger apoptosis [30,31]. When the UPR is activated, the glucose-regulated protein 78 (GRP78) unbinds three protein sensors in the ER that trigger parallel signalling pathways: inositol requiring enzyme 1 (IRE1), double-stranded RNA-activated protein kinase (PKR)-like ER kinase (PERK), and activating transcription factor 6 (ATF6) [32,33,34]. If ER stress cannot be resolved, the UPR switches from an adaptive survival mode towards the induction of apoptosis, often by modulating the expression of the proapoptotic CAAT/enhancer-binding protein homologous protein (CHOP), in turn leading to downregulation of antiapoptotic proteins, upregulation of proapoptotic proteins, and the initiation of caspase activity [35,36,37,38]. Carfilzomib has been shown to induce ER stress in several cancer types [39,40,41]. However, whether carfilzomib can induce ER stress and subsequent apoptosis in cetuximab-resistant colorectal cancer cell lines is unclear.

Here, we engineer two wt K-RAS colorectal cancer cell lines to generate acquired resistance to cetuximab. Subsequently, we screen a series of FDA-approved agents for their ability to inhibit cetuximab-resistant colorectal cancer cell lines. We identify carfilzomib as our lead candidate. Furthermore, we investigate the mode of mechanism of how carfilzomib can overcome the resistance caused by cetuximab treatment and discover that carfilzomib can initiate the UPR and mediate apoptosis.

## 2. Materials and Methods

**Cell culture and inhibitors:** Culture of human CRC cell lines was described previously [42,43]. All 12 cell lines had wt K-RAS and B-RAF expression [24]. All cells were maintained in Dulbecco’s Modified Eagle’s Medium (Life Technologies, Carlsbad, CA, USA) contained 5% foetal bovine serum (FBS) (Life Technologies), 100 U/mL penicillin, and 100 µg/mL streptomycin (Life Technologies). Cells were incubated in a humidified atmosphere of 90% air and 10% CO_2_ at 37 °C. Cetuximab was obtained from Merck KGaA (Darmstadt, Germany). The 38 drugs used in our screen (including carfilzomib) were all obtained from Selleck Chemicals (Houston, TX, USA).

**Generation of resistant cells:** LIM1215 and SW48 cells were cocultured with continuous, increasing doses of cetuximab for >4 months until treatment-selected populations of cells (designated LIM-CetR and SW-CetR) displayed resistance to concentrations of 20 µg/mL. Specifically, cells were cultured in an initial dose of 1 µg/mL of cetuximab with fresh media containing cetuximab added weekly. This dose of cetuximab was increased to 5 µg/mL, then 10 µg/mL, and then finally 20 µg/mL over the course of the 4-month treatment. Cell viability assays were performed to confirm whether cells were resistant after the 4-month coculture protocol.

**Cell-viability assay:** Cells were seeded in 96-well plates and allowed to adhere overnight. Triplicate wells were then treated with varying concentrations of inhibitors as indicated for 3–7 days. Cells were subsequently lysed and cell viability relative to the vehicle control was determined using a commercially available CellTiter-Glo kit (Promega) following manufacturer’s instructions. Samples were read on a bioluminometer.

**Wound-healing assay:** Cells were seeded into 6-well plates and were cultured until 100% confluent. After this, a wound was created with a p200 pipette tip. Cells were then treated with vehicle control, cetuximab (20 µg/mL), or carfilzomib (10 nM) and phase-contrast images were acquired at 0, 24, 48, and 72 h post-scratch. An inverted microscope (IX50, Olympus, USA) and the Leica Application Suite (LAS v4.5) were used to process and capture images. ImageJ was utilised to quantify wound closure.

**Colony-formation assay:** The colony formation assay was adapted from ‘clonogenic assay of cells in vitro’ by [44]. Cells were plated in 24-well plate at a cell density of 100 cells per well and allowed to adhere overnight. Triplicate wells were treated with vehicle control, cetuximab (20 µg/mL), or carfilzomib (10 nM) for 10–14 days at 37 °C and 10% CO_2_. After the treatment period, cells were washed and a mixture of 6.0% glutaraldehyde and 0.5% crystal violet was added for 30 min, following another wash, and then allowed to dry overnight. Quantification method of the colonies was adapted and utilised using ImageJ Plugin.

**Transwell-migration assay:** Cells were placed in the upper chamber of transwell plates (8 µm; Corning, NY, USA) in serum-free media while media containing FCS was added to the lower chamber. After 24 h, cells remaining on the upper side of the membrane were removed using a cotton swab. Cells on the underside of the membrane were then fixed in methanol, stained with crystal violet, and washed 3 times with PBS. The percentage of membrane covered by the cells was then calculated.

**RNA extraction and RT-PCR:** Cells were seeded in 6-well plates and allowed to adhere overnight. Following cell treatments and/or transfections, total RNA was extracted with the RNeasy Mini Kit (Qiagen; Hilden, Germany) following the manufacturer’s instructions. Reverse transcription was performed using the High-Capacity RNA-to-cDNA Kit (Applied Biosystems; Waltham, MA, USA). Reverse Transcription-PCR was performed using the GeneAmp PCR System 2400 (Perkin Elmer, Waltham, MA) under the conditions of 37 °C for 60 min and 95 °C for 5 min at a reaction volume of 20 µL. In order to quantify the transcripts of the genes of interest, real-time PCR was performed using the ViiA 7 Real-Time PCR System (Applied Biosystems) for CHOP/DDIT3: (Applied Biosystems, Hs00358796_g1), and GAPDH (Applied Biosystems, Hs02758991_g1). Amplified RNA samples were calculated using the 2^−∆∆CT^ method [45].

**ATF6-luc luciferase assay:** Cells were transfected with the ATF6 luciferase construct (pGL4.39; *Promega*) and allowed to adhere overnight. After 24 h, cells were then treated with 0 and 10 nM of carfilzomib for 24 h. Following another 24 h, cells were lysed and assessed for ATF6 luciferase activity with the use of the Luciferase Reporter Assay Kit (Promega) following the manufacturer’s instructions. Readings from lysed cells that were treated with control (i.e., without carfilzomib) were normalised to 1 and all subsequent readings were adjusted accordingly relative to control-treated readings.

**Caspase-3/7 assay:** Cells were plated in 96-well plates and allowed to adhere overnight. Triplicate wells were treated with 0 and 10 nM of carfilzomib for 24 h. Cells were then lysed and apoptosis was measured using the Caspase-Glo 3/7 Assay kit (Promega) following manufacturer’s instructions. Cell lysates were read on a bioluminometer.

**Subcutaneous xenograft animal model:** LIM1215, LIM-CetR, SW48, and SW-CetR (5 × 10^6^) cells were inoculated subcutaneously into both flanks of 6–8 weeks old BALB/c nude mice (Animal Research Centre, Western Australia, Australia). Tumour volume in mm^3^ was determined as previous [46]. For experiments involving cetuximab administration, mice were separated into 2 groups of five mice when tumours had reached approximately 100–150 mm^3^. Mice were subsequently treated 3 times weekly by i.p. injection at doses of 0 or 1 mg (~50 mg/kg) for 2–4 weeks. All animals were housed in cages (five mice per cage) in ambient temperatures for the duration of the experiment. The light cycle was controlled to provide 12 h light and 12 h darkness and humidity was approximately 40–60%. A standard diet of rodent pellets and tap water (membrane-filter purified and autoclaved) was provided ad libitum. This research project was approved by the Animal Ethics Committee of the University of Melbourne (Ethics agreement number 1613824).

## 3. Results

### 3.1. Generation and Characterisation of Acquired Cetuximab-Resistant Cell Lines

The LIM1215 and SW48 subpopulations with acquired resistance to cetuximab were generated through continuous coculturing with cetuximab over a period of 6 months, starting with low-dose concentrations (0.5–1 µg/mL), until cells were refractory to 20 µg/mL. These cetuximab-resistant subpopulations were designated LIM-CetR and SW-CetR and showed significantly greater in vitro resistance to cetuximab compared with their parental counterparts (Figure 1A). The IC_50_ for each cell line was LIM1215 = 4.52 µg/mL, LIM-CetR > 80 µg/mL, SW48 = 13.96 µg/mL, and SW-CetR > 80 µg/mL. In addition, cetuximab significantly reduced the wound-healing capacity and transwell migration of LIM1215 and SW48 parental cell lines but did not reduce the wound-healing capabilities of the LIM-CetR and SW-CetR cell lines (Figure 1B,C). Similarly, the cetuximab-resistant subpopulations showed significantly greater cell proliferation in vitro as measured by cell viability and colony formation (Figure 1D,E). Subcutaneous tumour xenograft analysis also showed that the LIM-CetR and SW48-CetR cells grew significantly faster compared with their parental counterparts (Figure 2A,B). As observed in our in vitro assays, cetuximab significantly reduced tumour growth of both LIM1215 and SW48 parental xenografts but did not have any significant effect on the LIM-CetR and SW-CetR xenografts (Figure 2C–F). Taken together the above data demonstrate that we have successfully generated subpopulations of cells with cetuximab acquired resistance.

### 3.2. Carfilzomib Inhibits Colorectal Cancer Cells with Acquired Resistance to Cetuximab

As we had generated cetuximab-resistant cell lines, we next explored the possibility of alternative therapeutics other than cetuximab to inhibit these cells. To do this we tested the efficacy of the standard dose of 1 µM of a set of 38 drugs consisting of chemotherapeutics and targeted agents using our cell viability assay as a readout of response on LIM-CetR and SW-CetR cells and their parental counterparts. The outcome of this screen is presented in Table 1 and demonstrates that carfilzomib was the most efficacious agent in our drug screen. Subsequent dose response studies indicated that carfilzomib had an IC50 of 5.3 nM in LIM-CetR cells and 4.0 nM in SW-CetR cells (Figure 3A). In addition, carfilzomib inhibited the colony formation in both LIM-CetR and SW-CetR cells to such an extent that no colonies were observed after 7 days of carfilzomib (10 µM) treatment (Figure 3B). As carfilzomib could significantly inhibit the proliferation of both LIM-CetR and SW-CetR cells, we evaluated whether carfilzomib could induce ER stress and apoptosis. To measure for the induction of ER stress, we tested for several markers of the unfolded protein response mechanism, specifically, CHOP expression and ATF6 activity. Indeed, carfilzomib could enhance the expression of CHOP (Figure 3C) and ATF6 activity (Figure 3D) in both LIM-CetR and SW-CetR cells. Finally, we assessed the level of caspase-3/7 activity in LIM-CetR and SW-CetR cells in the presence and absence of carfilzomib. Carfilzomib induced an approximately four-fold increase in caspase-3/7 activity in LIM-CetR and SW-CetR compared with vehicle control-treated cells (Figure 3E). Taken together, these data indicate that carfilzomib can reduce the proliferation of CRC cells with acquired resistance to cetuximab via the induction of the unfolded protein response and subsequent initiation of apoptosis.

**Carfilzomib inhibits colorectal cancer cells with intrinsic resistance to cetuximab.** As carfilzomib could inhibit the proliferation of CRC cells with acquired resistance to cetuximab, we next determined whether carfilzomib had similar effects on cells with intrinsic resistance to cetuximab. To identify wt K-Ras CRC cells with intrinsic resistance to cetuximab, we tested the efficacy of cetuximab on 12 wt K-RAS CRC cell lines [24]. Of these cell lines, cetuximab could inhibit six cell lines by greater than 50% after 3 days of treatment while six cell lines were relatively resistant to cetuximab (less than 50% inhibition) (Figure 4A). Based on these findings, we next tested carfilzomib on the four cell lines that displayed the greatest resistance to cetuximab (where no significant differences were seen in proliferation with or without cetuximab). Subsequently we treated these four cell lines (HCA-7, KM12, Colo320, and HT115), with carfilzomib (10 µM) for 3 days. Carfilzomib successfully inhibited the cell viability of all four of the cell lines tested.

## 4. Discussion

Cetuximab is only administered to mCRC patients with wt K-RAS and is only effective against some of these patients. Furthermore, patients who initially respond develop acquired resistance to cetuximab. The generation of cetuximab-refractory subpopulations of cells from cetuximab-sensitive parental cell lines through continuous exposure to cetuximab has been well established [47,48,49,50,51,52] and provides a robust model for evaluating novel resistance mechanisms and potential efficacious therapeutics. The two cetuximab-resistant cell lines we generated for this study (LIM-CetR and SW-CetR) showed enhanced proliferation either in the presence or absence of cetuximab compared with parental cells in both cell viability, colony formation assays, and in vivo xenograft growth. Napolitano et al. [53] made similar observations with their cetuximab-resistant cell lines indicating that cetuximab-resistant cells may have enhanced proliferative activity. In addition to an enhanced proliferative capability, our cetuximab resistant cells also showed an increased migratory effect as compared with parental cells (wound-healing and transwell migration assays) indicating a greater metastatic potential of these cells compared with their cetuximab-sensitive matched parental counterparts. Our results also corroborate other studies that looked at cetuximab-resistant cell lines. Although the cell lines used were different, others have found that resistant cell lines have stronger migratory capability [48].

The generation of these cell-line models for cetuximab resistance also allows for the development of robust, cost-effective screening of potential agents that may provide an alternative and improved treatment strategy to that of an ineffective cetuximab-based strategy. The rationale behind identifying novel agents for cetuximab-refractory patients can be based on either screening and potentially re-purposing a well-established drug that has been approved by the FDA, or assessing the efficacy of a new, unevaluated agent. We chose to test FDA-approved agents as we reasoned that testing “new” agents would result in a lengthy and expensive process to progress the agent through several rounds of pre-clinical and clinical evaluation before it could be potentially approved to treat patients with cetuximab-resistant wt K-RAS mCRC. Alternatively, repurposing FDA-approved agents should bypass many of these regulatory hurdles and fast-track the agent(s) for clinical evaluation. Ultimately, we screened a set of 38 drugs that consisted of chemotherapeutic and targeted agents using cell viability as our primary readout. The cetuximab-resistant cell lines showed varied responses to our drug panel, with most agents producing no significant effect on proliferation at the concentration chosen. As expected, cetuximab-resistant cells were also resistant to other inhibitors that block the EGFR family of tyrosine kinases, including afatinib, lapatinib, gefitinib, and erlotinib (although the dose of erlotinib used also had no significant effect on the LIM1215 and SW48 parental cell lines). Of the 38 drugs tested, carfilzomib was the most effective, with IC50 values in the low nM range.

Carfilzomib has been most used in treating multiple myeloma with great success [21]. Its use in solid tumour treatment is currently being studied. A proteasome inhibitor could potentially inhibit several cellular processes as the proteasome is vital in removing unwanted cellular debris as well as regulating specific key proteins [19]. Our analysis showed that at a very low concentration, carfilzomib was able to inhibit cell proliferation by more than 90% and significantly inhibited colony formation. Given that the primary function of the drug is to inhibit the proteasome, it could be hypothesised that an accumulation of cellular debris and potentially increased unfolded or misfolded proteins in the cell might cause toxicity, ER stress, and ultimately cell death. Indeed, carfilzomib has been shown to induce ER stress and the unfolded protein response (UPR) in other cancer settings. Lee et al. showed that the combination of bortezomib and carfilzomib led to increased expression in several UPR-related proteins and induction of apoptosis in melanoma cells [40]. Similarly to our current results, Forsythe et al. and Lamothe et al. demonstrated that carfilzomib could induce CHOP expression and cell death in several colorectal cancer cell lines and PBMCs from chronic lymphocytic leukaemia patients, respectively [25,39]. Our data also provide evidence that carfilzomib can provide antitumour activity via the induction of the UPR (enhanced CHOP expression and ATF6 activity) and the initiation of apoptosis (enhanced caspase-3/7 activity).

In summary, the present study characterises an in vitro model for cetuximab acquired resistance in colorectal cancer. Our current data indicate that these cetuximab-resistant clones proliferate, migrate, and grow as in vivo subcutaneous xenografts significantly faster than their cetuximab-sensitive parental counterparts. We also identify carfilzomib as a potential treatment option for intrinsic and acquired cetuximab-resistant colorectal cancer. Our current data support the potential clinical use of carfilzomib for the treatment of mCRC patients who harbour wt K-RAS expression and are refractory to cetuximab.

## Figures and Tables

**Figure 1 ijms-22-07114-f001:**
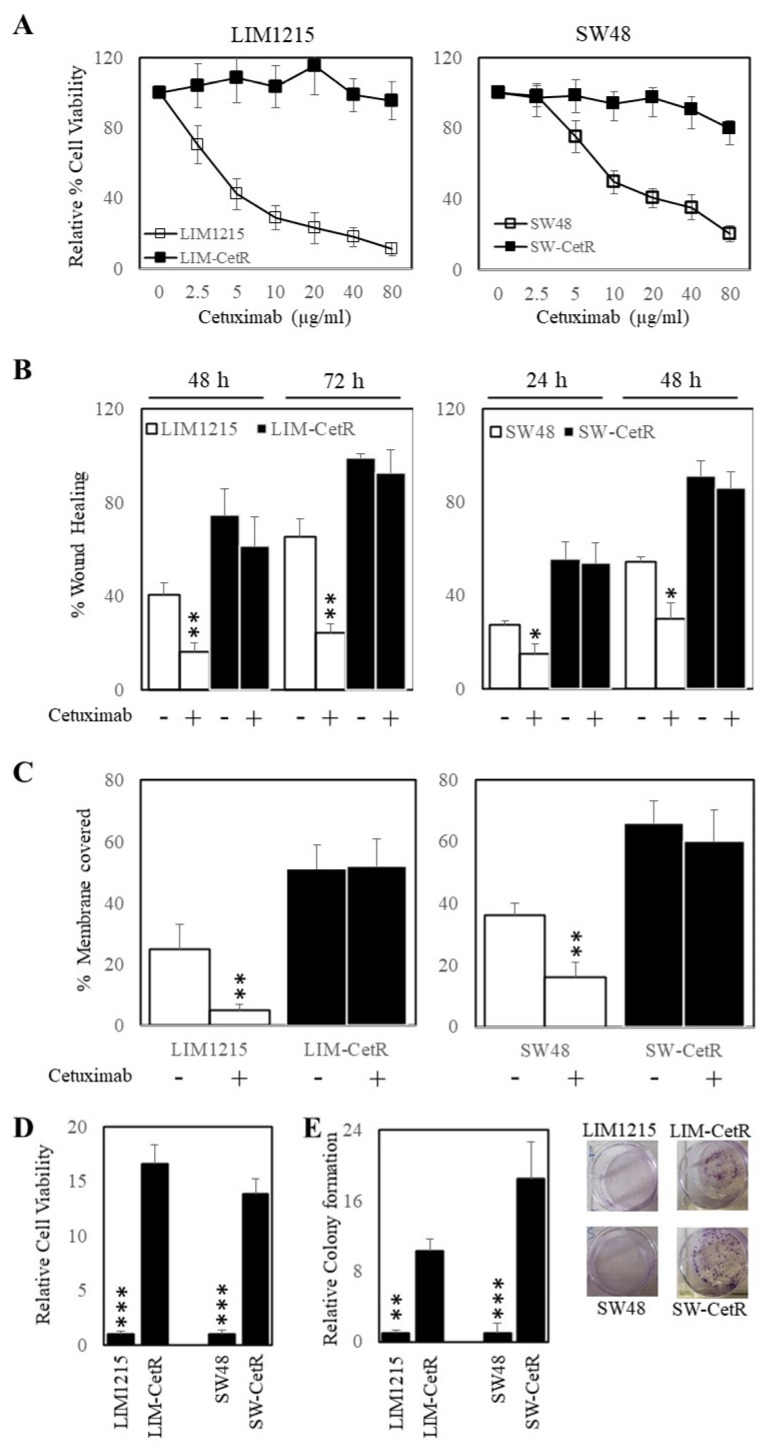
Cetuximab-acquired-resistant cells display greater proliferation. (**A**) LIM1215 (□) and LIM-CetR (■) cells (left-hand panel) and SW48 (□) and SW-CetR (■) cells (right-hand panel) were treated with a range of doses (0–80 µg/mL) of cetuximab for 72 h. Cell viability was then determined using a commercially available CellTiter-Glo kit and samples read on a bioluminometer. Data are expressed as % viability compared with untreated control cells ± SD. (**B**) LIM1215 (□) and LIM-CetR (■) cells (left-hand panel) and SW48 (□) and SW-CetR (■) cells (right-hand panel) were grown to confluency, then “wounded” at time 0 h. Cells were then treated with 0 and 20 µg/mL of cetuximab for 72 h. Images of wound healing were taken at 0, 24, 48, and 72 h post cetuximab treatment. Graphical representation of % wound remaining relative to control-treated cells at time 0 h. (**C**) LIM1215 (□) and LIM-CetR (■) cells (left-hand panel) and SW48 (□) and SW-CetR (■) cells (right-hand panel) were seeded to the upper chamber of a transwell plate and then treated with 0 and 20 µg/mL of cetuximab for 48 h. Cells on the underside of the membrane were stained with crystal violet and the % of membrane covered by cells was determined. (**D**) Cells were seeded and allowed to proliferate for 72 h. Cell viability was then determined as above. (**E**) Cells were seeded at a density of 100 cells per well and allowed to adhere overnight then treated with 0 and 20 µg/mL of cetuximab for 10–14 days. After the treatment period, colonies were counted using Image J. Data points represent mean ± SD of at least three independent experiments, each with three experimental replicates. * *p* < 0.05; ** *p* < 0.01; *** *p* < 0.001.

**Figure 2 ijms-22-07114-f002:**
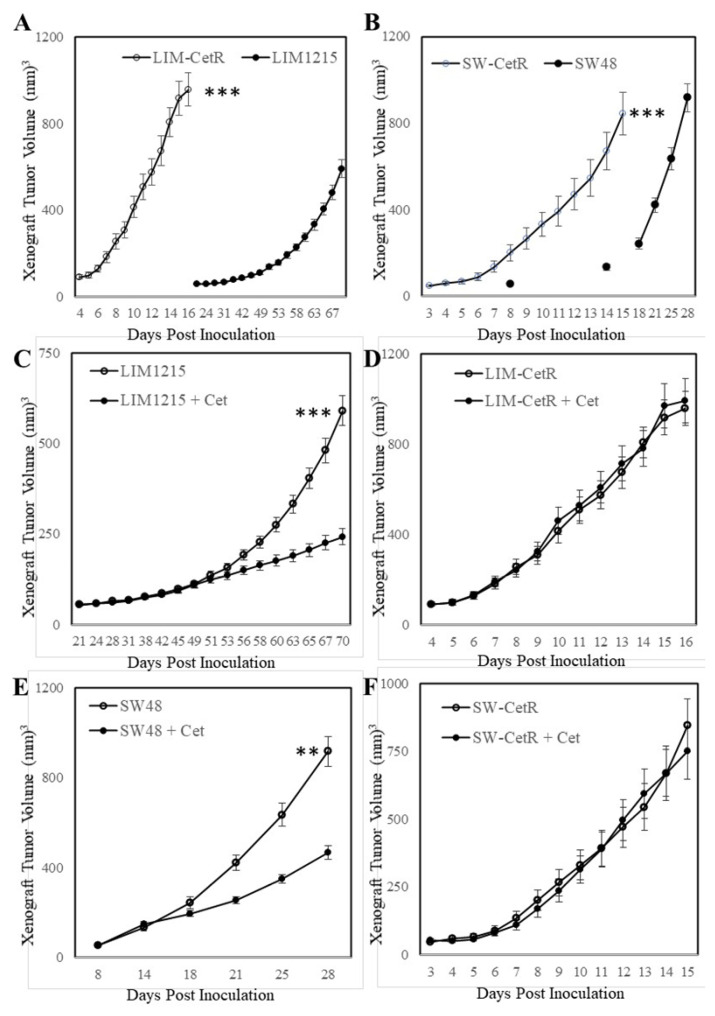
Cetuximab-acquired-resistant cells display greater in vivo growth. (**A**) LIM1215 and LIM-CetR cells and (**B**) SW48 and SW-CetR cells were subcutaneously injected into BALB/cnu/nu female mice. Data shown represent mean ± SEM (*n* = 10–12 tumours/group). (**C**) LIM1215, (**D**) LIM-CetR, (**E**) SW48, and (**F**) SW-CetR cells were subcutaneously injected into BALB/cnu/nu female mice. When mean tumour volume had reached 100–150 mm^3^, the mice were randomly separated into two groups and treated with vehicle (○) or cetuximab 3 times/week (●) for 3 to 4 weeks. Data shown represent mean ± SEM (*n* = 10–12 tumours/group). ** *p* < 0.01; *** *p* < 0.001.

**Figure 3 ijms-22-07114-f003:**
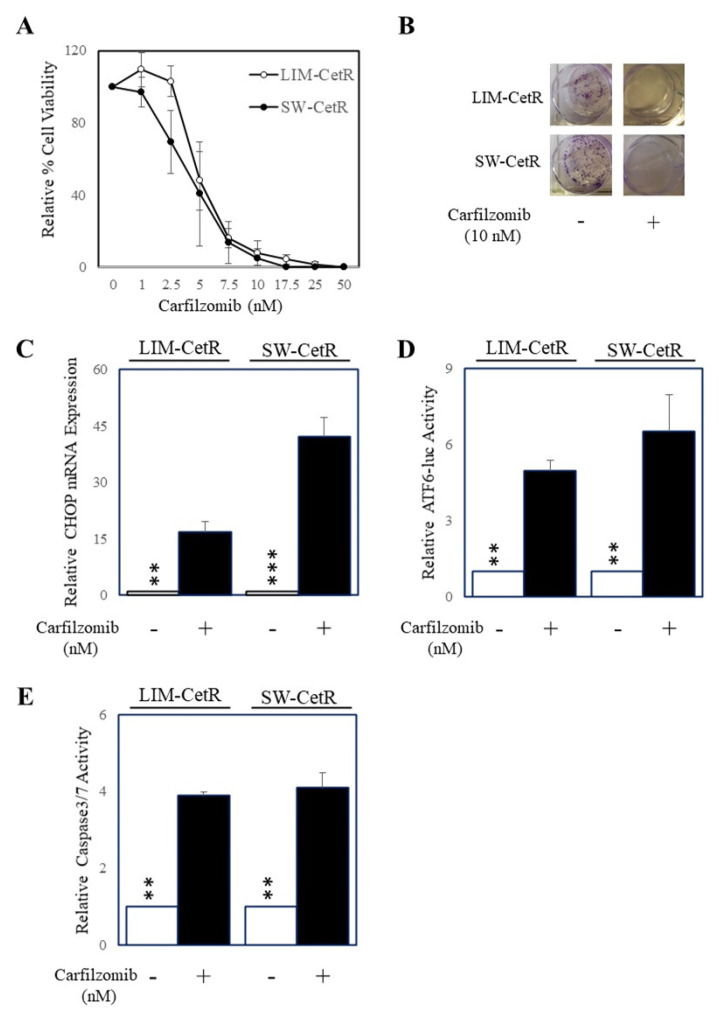
Carfilzomib inhibits the proliferation of cells with acquired cetuximab resistance and induces ER stress and apoptosis. (**A**) LIM-CetR (○) and SW-CetR cells (●) were treated with increasing doses of carfilzomib (0–50 nM) for 72 h. Cell viability was then determined using a commercially available CellTiter-Glo kit and samples read on a bioluminometer. Data are expressed as % viability compared with untreated control cells ± SD. (**B**) Cells were seeded at a density of 100 cells per well and allowed to adhere overnight then treated with 0 and 20 µg/mL of cetuximab for 10–14 days. After the treatment period, colonies were counted using Image J. (**C**) LIM-CetR and SW-CetR cells were treated with 0 or 10 nM of carfilzomib for 24 h then assessed for DDIT3 (CHOP) and GAPDH gene expression by qPCR. (**D**) LIM-CetR and SW-CetR cells were transfected with the ATF6 luciferase reporter construct and allowed to adhere overnight. Cells were then treated with 0 or 10 nM of carfilzomib for a further 24 h, lysed, and assessed for luciferase activity. Data are expressed as relative luciferase activity (fold change) by standardising the luciferase activity of the untreated cells to 1, and accordingly normalising all other raw values. (**E**) LIM-CetR and SW-CetR cells were seeded and allowed to adhere overnight. Cells were then treated with 0 or 10 nM of carfilzomib for a further 24 h, lysed, and assessed for caspase-3/7 activity. Data are expressed as relative caspase-3/7 activity (fold change) by standardising the caspase-3/7 activity of the untreated cells to 1, and accordingly normalising all other raw values. ** *p* < 0.01; *** *p* < 0.001.

**Figure 4 ijms-22-07114-f004:**
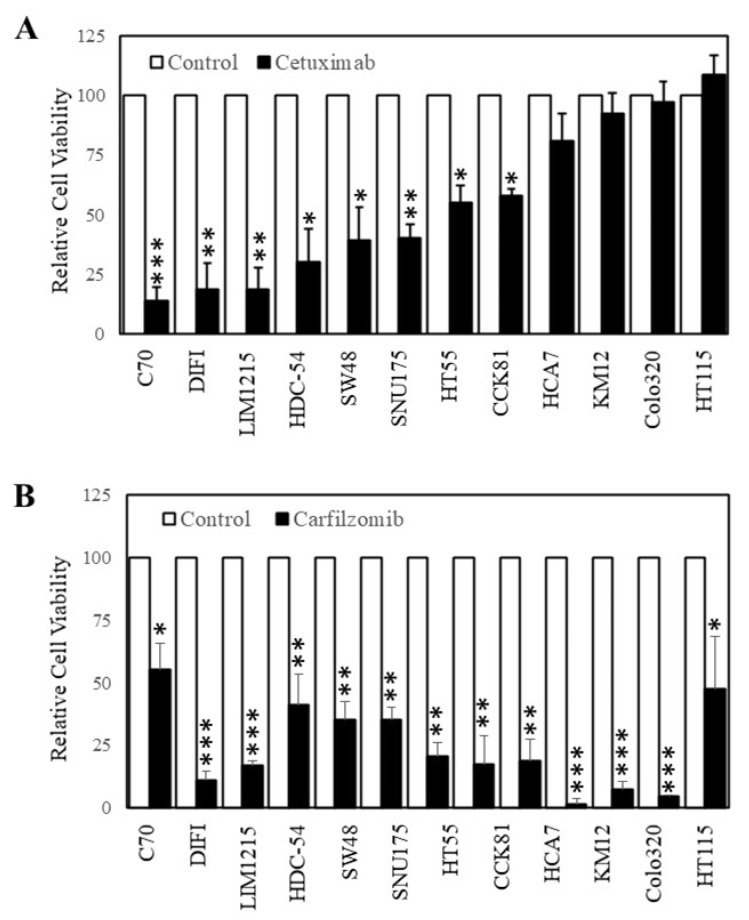
Carfilzomib inhibits the proliferation of colon cancer cells irrespective of their sensitivity to cetuximab. Cells were treated with (**A**) 0 (□) and (■) 20 µg/mL of cetuximab or (**B**) 0 (□) and (■) 10 nM of carfilzomib for 72 h. Cell viability was then determined using a commercially available CellTiter-Glo kit and samples read on a bioluminometer. Data are expressed as % viability compared with untreated control cells ± SD. * *p* < 0.05; ** *p* < 0.01; *** *p* < 0.001.

**Table 1 ijms-22-07114-t001:** Percentage proliferation of CRC cells after treatment with 38 therapeutic agents (1 µM).

Drug	LIM1215	LIM-CetR	SW48	SW-CetR
DMSO control	100.00	100.00	100.00	100.00
Acadesine	119.77	95.23	93.28	100.77
Afatinib	18.78	93.71	55.16	97.41
Aminosalicylate sodium	132.93	97.98	123.89	101.23
Apatinib	180.49	98.62	141.19	105.17
Axitinib	136.60	97.67	130.47	104.04
Azelnidipine	148.98	95.80	127.15	104.25
Bisoprolol	148.70	97.21	128.14	107.63
Bosutinib	150.12	94.05	89.78	114.99
Cabozantinib	168.01	96.34	132.71	101.71
**Carfilzomib**	**0.17**	**0.34**	**0.36**	**0.02**
Crizotinib	121.20	99.78	89.24	93.82
Dasatinib	66.63	72.63	38.00	55.49
Dexamethasone	116.76	100.73	103.43	97.36
Diclofenac	123.72	99.24	119.04	98.46
Enzalutamide	80.98	98.58	59.50	99.57
Erlotinib HCl	126.71	98.35	107.96	99.11
Everolimus	81.51	74.39	73.61	48.96
Evista	133.94	96.29	105.28	95.96
Gefitinib	61.67	86.68	55.08	88.92
Ibrutinib	23.89	95.84	52.37	96.63
Imatinib mesylate	117.71	100.22	117.19	96.93
Irinotecan	83.54	62.15	34.01	53.35
Lapatinib	32.16	96.55	66.56	97.60
Lenalidomide	130.12	97.45	123.61	91.34
Masitinib	131.84	95.78	93.71	89.78
Niclosamide	133.66	94.45	130.87	92.32
Nilotinib	183.86	82.47	191.23	88.46
Nystatin (Mycostatin)	126.32	93.01	154.25	92.84
Pazopanib HCl	91.72	80.10	80.85	88.57
Ponatinib	119.96	56.59	108.29	69.50
Regorafenib (BAY 73-4506)	107.05	92.17	98.12	88.42
Resveratrol	91.12	97.65	109.41	90.56
Ruxolitinib (INCB018424)	100.87	93.08	104.89	85.55
Temsirolimus (Torisel)	76.89	66.05	74.76	52.96
Tofacitinib citrate (CP-690550)	101.68	94.48	105.54	86.10
Vemurafenib	106.57	59.66	121.68	44.70
Vismodegib	92.23	85.23	112.89	73.74
Vorinostat (SAHA)	87.65	80.84	116.45	74.76

## Data Availability

Not applicable.

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
