# Peer review of "Carfilzomib Promotes the Unfolded Protein Response and Apoptosis in Cetuximab-Resistant Colorectal Cancer"

_ijms, 2021, doi:10.3390/ijms22137114_

Round 1

Reviewer 1 Report

  1. This manuscript is interesting and well-done.
  2. Theme of this research, to suggest a better therapeutic approach of cetuximab-resistant colorectal cancer. In this study, author proposed novel clinical approach by 38 current FDA approved agents included carfilzomib on cetuximab-resistant colorectal cancer and is well organized for readers to understand.
  3. Only minor points to be considered: English language should be revised throughout the text.
  4. It's just my opinion, ‘Introduction’ is too long and need to be rearrange. Please check IJMS guidelines and ensure that you comply with all the recommended guidelines. Present form is not the format required by this journal.
  5. 6p~7p, ‘Subcutaneous Xenograft Animal Model’, i.p. dose is 0, 1mg for 2-4 weeks. Is there ‘1 mg’ is 1mg/kg? How about add to information of mouse experimental environment? For example, ‘All animals were housed in cages (five mice per cage) for 2-4weeks. The light cycle was controlled to provide 12 h light and 12 h darkness; temperature was 22°C and humidity was 40–60%. A standard diet of rodent pellets and tap water (membrane-filter purified and autoclaved) were provided ad libitum’.
  6. It's just my opinion, 17p, STDEV is too high in figure 1A light panel (LIM 1215, cetuximab +, LIM-CetR , black bar) and figure 1D light panel (SW-CetR). Is there any specific reason for that?
  7. This paper is well organized for readers to understand. A main theme is so interesting too.

Author Response

We thank reviewer 1 for their thorough assessment of our manuscript and we welcome their suggested improvements. We have modified our manuscript in line with their comments as outlined below:

Reviewer 1, Comment 1: This manuscript is interesting and well-done.

Our comment 1: We thank the reviewer for this favourable comment. No changes to the manuscript have been made based on this comment.

Reviewer 1, Comment 2: Theme of this research, to suggest a better therapeutic approach of cetuximab-resistant colorectal cancer. In this study, author proposed novel clinical approach by 38 current FDA approved agents included carfilzomib on cetuximab-resistant colorectal cancer and is well organized for readers to understand.

Our comment 2: We thank the reviewer for this favourable comment. The reviewer has succinctly summarised our manuscript and has understood the key messages/findings we aimed to disseminate in this current manuscript. No changes to the manuscript have been made based on this comment.

Reviewer 1, Comment 3: Only minor points to be considered: English language should be revised throughout the text.

Our comment 3: We thank the reviewer for this excellent comment. We have carefully re-read the entire manuscript and have had an independent (non-author) read the manuscript for grammatical and typographic errors. Any changes to the manuscript based on these re-reads have been completed in the tracked changes version of the manuscript.

Reviewer 1, Comment 4: It's just my opinion, ‘Introduction’ is too long and need to be rearrange. Please check IJMS guidelines and ensure that you comply with all the recommended guidelines. Present form is not the format required by this journal.

Our comment 4: We thank the reviewer for their suggestion to improve our manuscript. We have shortened the introduction in some areas without losing the integrity and flow of our required background literature. These changes are clearly indicated in the tracked changes version of the manuscript.

Reviewer 1, Comment 5: 6p~7p, ‘Subcutaneous Xenograft Animal Model’, i.p. dose is 0, 1mg for 2-4 weeks. Is there ‘1 mg’ is 1mg/kg? How about add to information of mouse experimental environment? For example, ‘All animals were housed in cages (five mice per cage) for 2-4weeks. The light cycle was controlled to provide 12 h light and 12 h darkness; temperature was 22°C and humidity was 40–60%. A standard diet of rodent pellets and tap water (membrane-filter purified and autoclaved) were provided ad libitum’.

Our comment 5: Yes, we agree with the reviewer that the description of our in vivo experiments is mis-leading. As such we have improved/expanded this sub-section. To prevent the misunderstanding of the dose of cetuximab used we have included 1mg (approx. 50mg/kg) to the methods section and the figure legend.

In addition, we have also included the following “All animals were housed in cages (five mice per cage) in ambient temperatures for the duration of the experiment. The light cycle was controlled to provide 12 h light and 12 h darkness and humidity was approximately 40–60%. A standard diet of rodent pellets and tap water (membrane-filter purified and autoclaved) were provided ad libitum’ as indicated by the reviewer.

Reviewer 1, Comment 6: It's just my opinion, 17p, STDEV is too high in figure 1A light panel (LIM 1215, cetuximab +, LIM-CetR , black bar) and figure 1D light panel (SW-CetR). Is there any specific reason for that?

Our comment 6: We thank the reviewer for noticing this important point. We agree that the error bars outlined by the reviewer are particularly large. However, we have double checked our raw data and error bar calculations and these are the correct values displayed in the figure. We have replaced Fig 1A with a dose dependent curve (as requested by reviewer 2) and when repeating these experiments as part of our dose dependent experiments we did reduce the error bar in Fig 1A. We did not change Fig 1D. Nonetheless, the error bars do not affect the overall clear statistically significant differences we see when comparing the effect of cetuximab and the relative colony formation of these cells.

Reviewer 1, Comment 7: This paper is well organized for readers to understand. A main theme is so interesting too.

Our comment 7: We thank the reviewer for this favourable comment. We also express gratitude to the reviewer for the clear and thorough review which has ultimately improved our manuscript greatly. No changes to the manuscript have been made based on this comment.

Reviewer 2 Report

I have read the manuscript with the title “Carfilzomib Promotes the Unfolded Protein Response and Apoptosis in Cetuximab-Resistant Colorectal Cancer”.

In this study, researchers generate two Cetuximab-resistant human CRC cells. They found that Carfilzomib, one current FDA-approved agent, can efficiently overcome the two Cetuximab-resistant human CRC cells.

The following questions/comments need to be addressed.

Major points:

  1. It’s very important to test whether Carfilzomib can inhibit the Cetuximab-resistance cell (LIM-CetR and SW-CetR) growth using an in-vivo mouse model. In my opinion, authors can use the xenograft model which they used in this study to test since they already measure the cell growth of Cetuximab-resistance cells using this model.

  1. The most common method to confirm the generation of drug-resistant cell lines is to measure the IC50 values of resistant and parent cell lines to this drug through cytotoxicity Assays. Researchers usually show the curve of cell viability with different concentrations of drug to show the difference between parent cell line and resistant cell line. It’s necessary to conduct this experiment. In my opinion, this is important to include this result.

  1. In Fig 1B, this research found that cetuximab significantly reduced the wound healing capacity of LIM1215 and SW48 parental cell lines but not LIM-CetR and SW-CetR cell lines. It’s better to also test whether cetuximab significantly affects the migration of these cell lines (LIM1215, SW48, LIM-CetR, and SW-CetR) using transwell migration assay.

  1. Carfilzomib had an IC50 of 5.3 nM in LIM-CetR cells and 4.0 nM in SW-Cet-R cells in Fig 3A. Have researchers measure the IC50 of Carfilzomib on LIM1215 and SW48. If so, please also show this result.

  1. In Fig 4B, the authors measure the effect of Carfilzomib (10nM) on the 4 cell lines with intrinsic cetuximab resistance. It’s better that researchers also measure the effect of Carfilzomib (10nM) on the rest 8 cell lines.

Minor points:

  1. In Fig 3C, the label of LIM-CetR and SW-CetR should be added which will tell readers which columns represent LIM-CetR or SW-CetR (similar to Fig 3d, and 3E).
  2. In Fig 2C-2F, it’s more clear if the label of cetuximab treatment is used “+ Cet” instead of “ Cet”. For example, “SW48 Cet” can be changed to “SW48 + Cet”.

Author Response

We thank reviewer 2 for their thorough assessment of our manuscript and we welcome their suggested improvements. We have modified our manuscript in line with their comments as outlined below:

Reviewer 2 comment 1: It’s very important to test whether Carfilzomib can inhibit the Cetuximab-resistance cell (LIM-CetR and SW-CetR) growth using an in-vivo mouse model. In my opinion, authors can use the xenograft model which they used in this study to test since they already measure the cell growth of Cetuximab-resistance cells using this model.

Our comment 1: We thank the reviewer for this suggested important experiment. We did indeed test the efficacy of Carfilzomib on our Cetuximab resistant colon cancer cell lines. However, we did not show these results as surprisingly, Carfilzomib did not significantly reduce the tumour volume of our cells when grown as tumor xenografts. We suspect that this may have been a formulation issue. Nonetheless, we hope that this result does not diminish our overall findings presented in this manuscript.

Reviewer 2 comment 2: The most common method to confirm the generation of drug-resistant cell lines is to measure the IC50 values of resistant and parent cell lines to this drug through cytotoxicity Assays. Researchers usually show the curve of cell viability with different concentrations of drug to show the difference between parent cell line and resistant cell line. It’s necessary to conduct this experiment. In my opinion, this is important to include this result.

Our comment 2: We thank the reviewer for their comment and have amended our manuscript in accordance with their excellent recommendation. Fig 1A, which previously showed the differential of cetuximab efficacy at only one dose, has been replaced by curves showing the response of our parental vs resistant cells across a range of cetuximab doses. The IC50 is now also quoted in the revised manuscript. We also note that no or very little effect is seen on our resistant cell lines even at the highest cetuximab concentration used (80 µg/ml). The figure legend for figure 1 has also been adjusted to accommodate this change in the tracked changes version of the manuscript.

Reviewer 2 comment 3: In Fig 1B, this research found that cetuximab significantly reduced the wound healing capacity of LIM1215 and SW48 parental cell lines but not LIM-CetR and SW-CetR cell lines. It’s better to also test whether cetuximab significantly affects the migration of these cell lines (LIM1215, SW48, LIM-CetR, and SW-CetR) using transwell migration assay.

Our comment 3: We agree with the reviewer, that transwell assays will confirmed enhanced migration of the cetuximab resistant sub-populations in conjunction with the wound healing assays presented in the original manuscript. Therefore, we have performed these assays and have presented the data as Fig 1C (with the original Fig 1C and D, now being changed to Fig 1D and E). Indeed, the cetuximab resistant sub-populations of both LIM1215 and SW48 showed enhanced migration and this migration was not inhibited by cetuximab. To revise the manuscript in accordance with the changes made for this comment, we have included a Transwell Migration Assay sub-section in the methods, adjusted some text in the results describing Figure 1, added to the discussion and changed the figure legend to include this new data.

Reviewer 2 comment 4: Carfilzomib had an IC50 of 5.3 nM in LIM-CetR cells and 4.0 nM in SW-Cet-R cells in Fig 3A. Have researchers measure the IC50 of Carfilzomib on LIM1215 and SW48. If so, please also show this result.

Our comment 4: We thank the reviewer for this thoughtful comment. Unfortunately, we did not evaluate the dose dependant effects of Carfilzomib on the parental cell lines and thus did not calculate Carfilzomib’s IC50 against these cells. Our major focus was to find potential novel/repurposed inhibitors to cetuximab-resistant colorectal cancer cells. Nonetheless, as Table 1 shows, Carfilzomib had similar anti-proliferative effects on the parental and Cetuximab-resistant sub-populations at the high dose of 1 µM.

Reviewer 2 comment 5: In Fig 4B, the authors measure the effect of Carfilzomib (10nM) on the 4 cell lines with intrinsic cetuximab resistance. It’s better that researchers also measure the effect of Carfilzomib (10nM) on the rest 8 cell lines.

Our comment 5: As mentioned in our above comment (comment 4), we have focused on finding a novel inhibitor for cetuximab-resistant cells and therefore only tested Carfilzomib on cells with either acquired resistance to cetuximab (LIM-CetR and SW-CetR) or intrinsic resistance to cetuximab (HCA-7, KM12, COLO320 and HT115) and as such only presented these data in the original manuscript.  Nonetheless, as the reviewer has made the comment to test Carfilzomib on the other 8 cell lines, we performed these studies and modified the Figure (Fig 4B). Importantly, Carfilzomib inhibited all 12 lines tested with no distinguishing difference between the cetuximab-sensitive and cetuximab-resistant cell lines. We have made the following changes to the manuscript based on this modification. We modified the results section describing Fig 4B and the figure legend of Figure 4.

Reviewer 2 comment 6:  In Fig 3C, the label of LIM-CetR and SW-CetR should be added which will tell readers which columns represent LIM-CetR or SW-CetR (similar to Fig 3d, and 3E).

Our comment 6: We thank the reviewer for this comment. We have made these changes. (see tracked changes in the revised manuscript).

Reviewer 2 comment 7: In Fig 2C-2F, it’s more clear if the label of cetuximab treatment is used “+ Cet” instead of “ Cet”. For example, “SW48 Cet” can be changed to “SW48 + Cet”.

Our comment 7: We thank the reviewer for this comment. We have made these changes (see tracked changes in the revised manuscript).

Round 2

Reviewer 2 Report

The authors have adequately addressed my concerns/questions. I don't have any more questions.